# Epidermal Nevus Syndrome Associated with Dwarfism and Atopic Dermatitis

**DOI:** 10.3390/children8080697

**Published:** 2021-08-12

**Authors:** Yuki Mizutani, Miki Nagai, Hitoshi Iwata, Kunihiro Matsunami, Mariko Seishima

**Affiliations:** 1Department of Dermatology, Gifu Prefectural General Medical Center, Gifu 500-8717, Japan; mknagai.nagai@gmail.com; 2Department of Pathology, Gifu Prefectural General Medical Center, Gifu 500-8717, Japan; 33136@gifu-hp.jp; 3Department of Pediatrics, Gifu Prefectural General Medical Center, Gifu 500-8717, Japan; kmatsunami@gifu-hp.jp; 4Department of Dermatology, Gifu University Graduate School of Medicine, Gifu 501-1194, Japan; marikoseishima@yahoo.co.jp

**Keywords:** epidermal nevus syndrome, FGFR3, Garcia–Hafner–Happle syndrome, dwarfism, atopic dermatitis

## Abstract

Epidermal nevus syndrome (ENS) is a congenital disorder characterized by widespread linear epidermal lesions consisting of epidermal nevus and extracutaneous involvements, especially of the central nervous system and skeletal system. Garcia–Hafner–Happle syndrome, also known as fibroblast growth factor receptor 3 (FGFR3)-ENS, is characterized by a systematized keratinocytic EN of soft and velvety type with neurological abnormalities such as seizures, intellectual impairment, and cortical atrophy. We present a case of a 9-year-old Japanese boy afflicted with Garcia–Hafner–Happle syndrome associated with dwarfism and atopic dermatitis. We show the results of physical examination, DNA analysis, and imaging studies and discuss the mutation underlying the child’s disorder.

## 1. Introduction

ENS represents a rare neurocutaneous disorder in which epidermal nevi are associated with abnormalities in extracutaneous organ systems (e.g., the central nervous system, cardiovascular system, genitourinary system, eyes, and bone) [1]. Happle et al. classified ENS by the types of associated epidermal nevus and also by the criterion of presence or absence of heritability. Well defined syndromes in hereditary skin diseases, including Schimmelpenning syndrome, phacomatosis pigmentokeratotica, nevus comedonicus syndrome, and Becker nevus syndrome, are characterized by organoid epidermal nevi. Three phenotypes with a known molecular etiology include congenital hemidysplasia with ichthyosiform nevus and limb defects (CHILD) syndrome, type 2 segmental Cowden disease, and FGFR3 epidermal nevus syndrome (FGFR3-ENS). FGFR3-ENS is characterized by the systemic appearance of soft, velvety, keratinocytic epidermal nevi, and neurological abnormalities such as seizures, intellectual impairment, and cortical atrophy. Although the cutaneous manifestation of FGFR3-ENS is similar to type 2 segmental Cowden disease, we could distinguish these two phenotypes only by genetic mutation [1]. Herein, we report the first case of a child with ENS who also suffered from dwarfism and atopic dermatitis.

## 2. Case Report

A 9-year-old Japanese boy was referred to us for brownish papules in the perianal region. He was born at 36 weeks and 6 days of pregnancy with short-limbed dwarfism that had been diagnosed in utero. He had a medical history of infantile respiratory distress syndrome, cerebral atrophy, seizure, autism, and atopic dermatitis (AD) without remarkable family history. Brown macules on all four extremities were first noted at 2 months; they spread to the buttocks and neck in a linear distribution pattern. Physical examination revealed widespread brown macules and white plaques along Blaschko’s lines, mammillated lesions on the lips, and brownish papillomatous plaques in the perianal region (Figure 1a–c). In addition, the patient had atopic dry skin on the whole his body and slightly pruritic lichenification on the four extremities. Histopathological examination of plaques in the perianal region showed papillomatous epidermal hyperplasia associated with elongated rete ridges and variable pigmentation in the basal layer of the epidermis. There was perivascular lymphocytic infiltration into the superficial dermis but no abnormalities in adnexal structures (Figure 2a–c). Electron microscopy revealed hyperkeratosis, deposition of intercellular lipoid cells, and normal cornified cell envelope (Figure 3a–c). Cerebral magnetic resonance imaging showed atrophy of the cerebral hemisphere. After obtaining informed parental consent, we extracted DNA directly extracted from the child’s blood sample, scalp hair roots, and a nail. A mosaic form of an R248C FGFR3 mutation (rs121913482) was identified in scalp hair roots and blood leukocytes. From these results, we diagnosed the patient with ENS caused by mosaicism of the FGFR3 mutation: Garcia–Hafner–Happle syndrome. One year of topical treatment with maxacalcitol and urea cream did not result in much improvement. Ablative carbon dioxide laser treatment is now the next line of treatment for his skin lesions.

## 3. Discussion

The FGFR3 gene plays an important role in regulating activities such as cell division, migration, and differentiation. FGFR3 protein is a negative regulator of bone growth, so gain-of-function point mutations in the germline can downregulate long-bone growth. It follows that activating germline mutations in FGFR3 induces dwarfism, skeletal dysplasia, and craniosynostosis syndromes [2]. FGFR3 mutation–induced dwarfism and craniosynostosis syndromes are sometimes associated with acanthosis nigricans (AN), a skin disorder that is clinically and histologically similar to epidermal nevi. AN is a brown-to-black velvety hyperpigmentation of the skin, mostly located at intertriginous area. From recent cases reports, FGFR3 mutation associated with AN resulted in less severe skeletal dysplasias. FGFR3 mutations are involved in the pathogenesis of a variety of malignant tumors, such as prostate, colon, cervical, and bladder carcinomas [3]. FGFR3 mutations have been studies as germline mutations in patients with AN and as somatic mutations in patients with seborrheic keratoses and epidermal nevi [3]. These cutaneous disorders have similar clinical and histological characteristics, which are thought to originate from proliferating keratiocytes and characterized by acanthosis, hyperkeratosis, and papillomatosis. The survival of patients with ENS is explained by mosaicism. Activating FGFR3 mutations are present in 41% of patients with epidermal nevi; moreover, they are exclusively found in epidermal nevi cases with R248C FGFR3 mutations (94%) [4].

Garcia–Hafner–Happle syndrome (also known as FGFR3-ENS) is caused by a mosaic R248C mutation in the FGFR3 gene. It is characterized by the systemic appearance of soft, velvety, keratinocytic epidermal nevi and neurological abnormalities such as seizures, intellectual impairment, and cortical atrophy [5]. Although R248C FGFR3 mutations have been reported in patients with epidermal nevi and its extracutaneous manifestations, this is the first reported case of ENS-associated dwarfism and AD as far as we know [6,7]. AD is characterized by persistent itching and eczematous lesions. Disruption of the skin barrier is an important pathogenesis of AD, which activates keratinocytes and polarizes attracted T cells into Th 2 cells, which secret IL-4 and IL-13, resulting in decreased expression of barrier-related gene products such as filaggrin [8]. Filaggrin loss-of-function variants are the most widely replicated genetic risk factor of AD [9]. However, filaggrin gene variants have been detected in only 10~30% of AD patients [10]. We assume that FGFR3 mutations could be involved in pathogenesis of AD in terms of keratinocyte activation via filaggrin loss-of-function. The lesions of AD are often variable and are formed typical with age. There are possibilities that the lesions of AD become typical with age. Thus, the following observation is necessary to clarify the association of FGFR3-ENS with AD.

Current studies indicate that the dwarfism and cerebral involvement observed here could be the result of an R248C FGFR3 mutation. As FGFR3 mutations are involved in the pathogenesis of malignant tumors, clinicians should monitor for the development tumors in patients with ENS who carry those mutations.

## Figures and Tables

**Figure 1 children-08-00697-f001:**
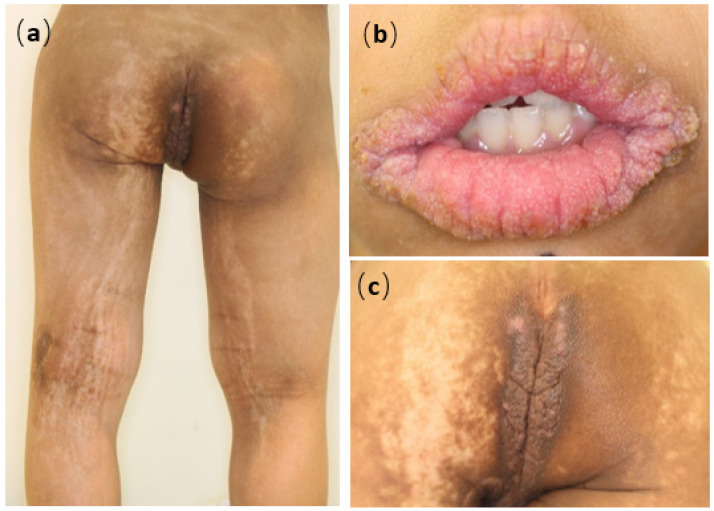
Clinical features of this patient: widespread brown macules and white plaques along Blaschko’s lines (**a**) with mammillated lesions of the lip (**b**) and brownish papillomatous plaques in the perianal region (**c**).

**Figure 2 children-08-00697-f002:**
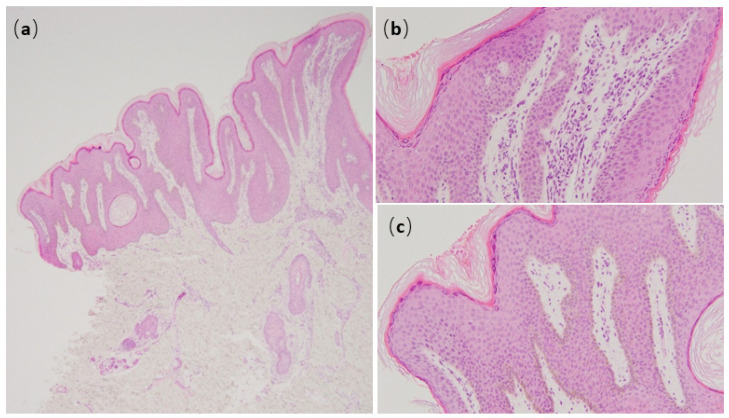
Histopathology of papillomatous plaques in the perianal region shows papillomatous epidermal hyperplasia associated with elongated rete ridges (**a**), perivascular infiltration of lymphocytes into the superficial dermis (**b**), and pigmentation of basal layer of epidermis (**c**).

**Figure 3 children-08-00697-f003:**
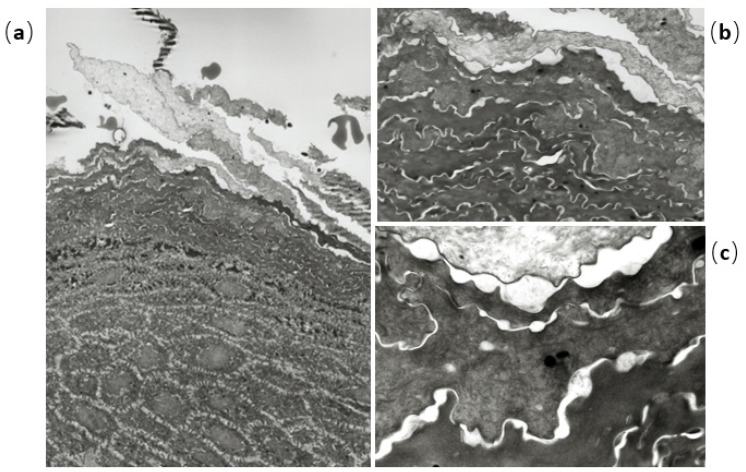
Electron microscopic findings show hyperkeratosis (**a**), deposition of intercellular lipoid cells (**b**), and normal cornified cell envelope (**c**).

## Data Availability

The data presented in this study are available upon request from the corresponding author. The data are not publicly available due to privacy restrictions.

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
