# Peer review of "Epidermal Nevus Syndrome Associated with Dwarfism and Atopic Dermatitis"

_children, 2021, doi:10.3390/children8080697_

Round 1

Reviewer 1 Report

Dear Authors.

The current manuscript describing the ENS caused by mosaicism of the FGFR3 mutation - Garcia–Hafner–Happle syndrome - in a 6 years-old boy seems to be an important description for Rare disease field.

During my review, I realise that simple and important information could improve your manuscript.

  • False Positive of Mosaicism diagnosis, sometimes, are due to methodology artefacts. Please assure for future readers that the methodology used was correct applied, given more detail about it.
  • The interesting findings was the atopic dermatitis, but lesion detail must be better explored and investigated.  Moreover, the hypothesis was that "We assume that FGFR3 mutations could be involved in pathogenesis of AD in terms of keratinocyte activation via filaggrin loss-of-function". Why others patients carrying the same FGFR3 mutation never present atopic dermatitis? Being this the first report?

In addition:

  1. Please insert the rsID for the FGFR3 mutation.
  2. Figure 2 and 3 are misspelling.

Regards

Author Response

I greatly appreciate your critical comments.

  1. We received the results of genetic testing, but unfortunately, we could not obtain the detail methods of genetic testing because the test was performed at the hospital where the patient was born 12 years ago.
  2. The lesions of atopic dermatitis were spread to the whole body. I added the detail of the lesions to page 2, line 51-52. Filaggrin loss-of-function variants are the most widely replicated genetic risk factor of AD, but all of AD patients do not have filaggrin loss-of-function variants. This point was added to page 4, line 147-148 with a reference (Reference 10). This is the first case of FGFR3-ENS with AD as far as we searched. We emphasized this point in page 1, line 37, and page 3, line 133-134. There w

The lesions of AD are often variable and are formed typical with age. There are possibilities that the lesions of AD become typical with age. Thus, the following observation is necessary to clarify the association of FGFR3-ENS with AD. This point was added to page 4, line 150-153.

  1. We inserted rsID for the FGFR3 mutation in page 2, line 60 (rs121913482).
  2. We corrected misspelling of Figure 2 and 3. I appreciated indicating the misspelling.

Reviewer 2 Report

Interesting case report with educational value.

Author Response

I greatly appreciate your critical comments.

Reviewer 3 Report

The authors have presented a very interesting case report of a rare syndrome. The case was correctly, broadly diagnosed. The article is well illustrated with photographs of histopathological examinations. 
I suggest the introduction of minor revisions:

I would suggest more extensive discussion of the differential diagnosis of Garcia-Hafner- Happle syndrome.

Author Response

 We appreciate your advice. FGFR3-ENS is characterized by the systemic appearance of soft, velvety, keratinocytic epidermal nevi and neurological abnormalities such as seizures, intellectual impairment, and cortical atrophy. Although the cutaneous manifestation of FGFR3-ENS is similar to type 2 segmental Cowden disease, we could distinguish these two phenotypes only by genetic mutation. We added this point to page 1, line 34-36.

Round 2

Reviewer 1 Report

Dear Authors,

Thank you to the response of my question and I appreciated the opportunity to follow this case.

Regards